# Micropropagation of *Ajuga bracteosa* via Direct Organogenesis Using Internodal Explants: SEM, GC-MS, and SCoT Marker Analysis

**DOI:** 10.3390/plants14162507

**Published:** 2025-08-12

**Authors:** Arisha Kausar, Anwar Shahzad, Aashiq Yousuf Bhat, Muthusamy Ramakrishnan, Zishan Ahmad

**Affiliations:** 1Plant Biotechnology Section, Department of Botany, Aligarh Muslim University, Aligarh 202002, India; arishakausar98@gmail.com (A.K.); bhataashiq@gmail.com (A.Y.B.); 2State Key Laboratory of Tree Genetics and Breeding, Co-Innovation Center for Sustainable Forestry in Southern China, Bamboo Research Institute, School of Life Sciences, Nanjing Forestry University, Nanjing 210037, China; ramky@njfu.edu.cn

**Keywords:** *Ajuga bracteosa*, medicinal plant, conservation, genetic fidelity, phytochemical profiling

## Abstract

*Ajuga bracteosa* is a herb with high medicinal value and a low range of distribution. It is used in several herbal and traditional medicines, including diabetes. In the present study, we designed the methodology for the micropropagation of *A. bracteosa* from internodal segments. The highest shoot multiplication was achieved on Murashige and Skoog (MS) medium supplemented with 6-benzyl-amino-purine (BAP) (5.0 µM) + indole-3-acetic acid (IAA) (1.5 µM) + adenine sulphate (ADS) (15.0 µM), which produced the maximum number of 20.45 ± 0.12 shoots/explants with 6.43 ± 0.006 cm shoot length. Rooting in the microshoots was attained on half-strength MS medium containing indole-3-butyric acid (IBA) (1.5 µM), with the highest root number of 16.44 ± 0.015 roots/shoot, and root length of 2.25 ± 0.011 cm. To assess genetic fidelity, SCoT marker analysis was performed on nine randomly selected *in vitro* regenerated plantlets and the mother plant, all of which exhibited monomorphic banding patterns, confirming genetic stability. Scanning electron microscopy (SEM) reveals normal stomatal structure in the regenerated plants post-acclimatization, indicating successful physiological recovery. Furthermore, Gas Chromatography–Mass Spectrometry (GC-MS) analysis confirms the presence of major phytocompounds in both the *in vitro* regenerated plants and the mother plant, supporting the conservation of phytochemical integrity. Given the restricted distribution and overharvesting pressure on this species, the established protocol provides an efficient strategy for rapid, large-scale, and genetically stable propagation to support conservation and pharmaceutical utilization.

## 1. Introduction

The use of medicinal plants has become prevalent in modern medicine. Globally, people are becoming more interested in adopting natural products rather than synthetic pharmaceuticals since they have fewer side effects than synthetic chemicals. However, these chemicals can have negative impacts [1]. The majority of tribal people use medicinal plants and their extracts to treat both serious illnesses, like cancer, cardiovascular, and Alzheimer’s disease, as well as minor ailments, like the common cold, cough, fever, and wounds [2,3,4,5,6]. Integrated multi-omics approaches have advanced our understanding of specialized metabolite biosynthesis in medicinal plants, revealing key regulatory networks and cell-specific dynamics. Despite progress, challenges in data management, standardization, and synthetic biology application remain [7]. *Ajuga bracteosa* Wall ex. Benth (*Labiatae*), also known as *neelkanthi*, is a herb that grows on slopes and even in rock crevices throughout the Himalayas, Meghalaya, Bihar, and Tamil Nadu at altitudes of 300–2400 meters [8]. It is a perennial plant that grows upright, reaching a height of 5 to 50 cm. Its leaves are often prostrate and have oblanceolate or sub-spathulate shapes. The herb is recommended in Ayurveda for the treatment of amenorrhea, rheumatism, palsy, and gout [9]. The leaves of the plant have localized benefits for treating headache, acne, measles, stomach acidity, jaundice, hypertension, sore throat, burns, and blood purification [10]. *A. bracteosa* has expectorant, anti-inflammatory, and cytotoxic potential [11,12]. Most importantly, the leaves are employed as a quinine alternative in the treatment of fevers. Additionally, the leaf extract of the plant displays anti-parasitic, insecticidal, cytotoxic, and anti-Alzheimer activity [13].

The market for *A. bracteosa* has grown significantly over the last ten years, both on the Asian subcontinent and in other nations [14]. Natural habitats of the species have been modified or destroyed due to overuse for local purposes and unregulated development activities, such as the construction of buildings, roads, and bridges, which could limit future distribution of the species. Additionally, landslides pose a secondary major threat. If these harmful factors persist, the species could face the risk of becoming threatened in the near future [15]. Medicinal plants of interest can be quickly multiplied *in vitro* through the use of micropropagation, a widely used plant tissue culture method [16,17,18]. Plant tissue culture is an efficacious way for metabolite production with a faster rate of growth and active secondary metabolism. In addition to rapid propagation and metabolite production, plant tissue culture plays a crucial role in the conservation of valuable and often endangered plant species, such as *A. bracteosa* [19,20]. When clonal propagation is needed, the benefit of plant tissue culture may be severely limited if obscured genetic deficits arise from somaclonal variation in regenerated plants. Therefore, to guarantee the quality of plantlets for their commercial worth, it is crucial to establish the genetic homogeneity of tissue culture-raised plants. Moreover, recent studies have revealed that N6-methyladenosine (m6A), a key RNA modification, plays a crucial role in regulating plant development and stress responses, potentially influencing tissue culture outcomes and morphological traits such as leaf formation and acclimatization efficiency [21,22]. Since stomata are primarily responsible for gas exchange and the transport of water and minerals to plants, their quantity will have an impact on plant growth. A comparison of the leaf morphology of *in vitro*-grown plants, as well as fully acclimatized plants, as well as the mother plant was investigated using scanning electron microscopy (SEM).

Micropropagation of *A. bracteosa* has previously been achieved through organogenesis using various explants, including nodes, petioles, leaves, and somatic embryos [23]. However, to date, there are no documented studies reporting direct organogenesis from internodal segments as explants in *A. bracteosa*. We also aimed to determine the phytoconstituents of *A. bracteosa* using GC-MS analysis. This is the first report using SCoT primers for genetic fidelity analysis of *in vitro*-grown plants and a mother plant and the first paper on the comparison of the leaf morphology of *in vitro*-grown plants and fully acclimatized plants of *A. bracteosa*. We also compared the physiological parameters of the plants as they progressed towards acclimatization.

## 2. Results and Discussion

### 2.1. Micropropagation

Murashige and Skoog (MS) [24] basal medium without PGRs did not exhibit any response in morphology, while different cytokinin concentrations exhibited their responses after 1 week of inoculation. In our study, while the internodal explants remained green and visibly healthy without the addition of growth regulators, their inability to produce shoots suggests that external phytohormones are essential for initiating bud break and promoting shoot development in these explants. Notably, Burbulis et al. also reported that internodes of *Passiflora edulis* failed to initiate shoot or root primordia on MS basal medium without the addition of exogenous plant growth regulators (PGRs) [25]. Dhir and Shekhawat also noted that the absence of exogenous growth regulators from the culture medium resulted in a complete lack of shoot induction in *Ceropegia bulbosa*, despite the explants maintaining a healthy and green appearance [26,27]. Among the different cytokinins tested, 6-benzyl-amino-purine (BAP) (5.0 µM) was discovered as the most efficacious in direct shoot regeneration, with 9.98 ± 0.057 shoots/explants with 2.60 ± 0.057 cm mean shoot length, exhibiting 94% ± 0.57 response (Table 1; Figure 1). This high performance of BAP aligns with previous findings that attribute its effectiveness to the enhanced ability of the plant tissue to metabolize naturally occurring hormones compared to synthetic regulators. Additionally, BAP may act indirectly by stimulating the endogenous production of natural cytokinins such as zeatin, thereby functioning through intrinsic hormonal pathways [28,29]. Moreover, kinetin (Kn) and 2-isopentanyladenine (2iP) (5.0 µM) also displayed effective responses, producing 5.15 ± 0.017 and 5.89 ± 0.005 shoots/explants, along with 1.70 ± 0.100 cm and 2.16 ± 0.088 cm mean length of shoots exhibiting a 65% ± 2.30 and 80% ± 0.57 response, respectively. Direct shoot induction in stem explants of *A. bracteosa* without callus formation and somatic embryogenesis was also observed in another paper [23]. BAP also demonstrated the highest efficiency in promoting axillary shoot regeneration in *Ajuga lupulina* [30]. However, we observed that higher concentrations of BAP (7.5 µM) resulted in the browning of leaves, as well as thin and weak shoot development. However, at low concentrations, BAP has been shown to enhance microshoot initiation with high shoot number and length [31]. Numerous studies supported our findings and demonstrated the effectiveness of the cytokinin BAP on shoot initiation in a range of herbaceous, woody, and resistant plants [32,33,34,35,36]. Additionally, BAP significantly influences the formation of foliar growth in shoot tip cultures, as well as the growth of adventitious and axillary buds [37,38]. As a result of increased cellular absorption and plasma membrane permeability, BAP is effective in the developmental processes of the plant, such as direct, as well as indirect, somatic embryogenesis, callus formation, bud breaking, efficient proliferation, and enhanced shoot multiplication [39,40,41,42,43]. However, elevated levels of BAP can lead to adverse effects, including reduced bud induction, reduction in shoot number and shoot length, and increased callus formation, which may compromise the regenerative capacity of the explants [44].

Organogenesis represents a highly regulated morphogenetic event characterized by the development of monopolar structures such as shoots or roots. In many cases, *in vitro* shoot organogenesis is stimulated by the synergistic interaction of cytokinins and auxins [45]. Notably, shoot regeneration generally requires a higher concentration of cytokinins compared to auxins [46]. BAP (5.0 µM), along with different concentrations of α-naphthalene acetic acid (NAA), indole-3-acetic acid (IAA), and indole-3-butyric acid (IBA), was found effective for shoot proliferation. BAP 5.0 µM + IAA 1.5 µM was found to give the best response, with 12.25 ± 0.028 shoots/explants and 4.19 ± 0.035 cm mean shoot length, providing 100% response in the cultures, after 4 weeks (Table 2). Similarly, Jan et al. also obtained good results in *A. bracteosa* on MS medium combined with BAP + IAA [47]. At the molecular level, the increased shoot production observed with BA-IAA treatments compared to treatments using only BA or IAA may be due to the combined influence of both hormones in activating cyclin-dependent kinases (CDKs), which are the key protein kinases that control the mitotic division rate through gene transcription in eukaryotic cells. This combined action likely enhances CDK expression at different phases of the mitotic cell cycle, indicating a synergistic effect that supports multiple shoot development [48,49]. In our study, a combination of BAP 5.0 µM + NAA 1.5 µM was also found effective in shoot regeneration, with a 12.15 ± 0.014 mean shoot number and a mean shoot length of 3.79 ± 0.017 cm. In comparison to GA_3_ and other auxins, the BAP + NAA combination worked best for shoot multiplication in *A. bracteosa* [50]. When BAP + NAA (3 + 1 mg L^−1^) was added to the medium, the shoots with 100% induction were observed, yielding 13.8 shoots/explants with 4.1 cm shoot length [50]. However, they failed to regenerate any shoots on MS medium containing BAP + IBA. We also observed that BAP 5.0 µM + IBA 1.5 µM exhibited the lowest shoot regeneration compared to other auxins, with 9.54 ± 0.012 shoots/explants and 2.93 ± 0.017 cm mean shoot length. The shoots obtained by combining BAP 5.0 µM and auxins (IBA, NAA, and IAA) showed fully expanded leaves and expedited length of the shoots in the culture. Auxins are essential to preserving the development of stem cells and controlling cell division, whereas cytokinins have been shown to regulate the rate of meristematic cell differentiation [51,52]. An optimal response is impacted if the frequency of stem cell generation is somewhat lower than the frequency of cell differentiation at an optimized cytokinin concentration. The current report is consistent with several studies that revealed that the combination of optimized auxin and cytokinin was more productive in terms of increased shoot output than cytokinin treatment alone [53,54,55].

For the further elongation and multiplication of shoots, an experiment was designed that contained BAP 5.0 µM + IAA 1.5 µM, along with different concentrations of adenine sulphate (ADS) and glutamine (GLU) (Table 3). The addition of ADS substantially enhanced the shoot number and shoot length in the cultures. In comparison to Kn, BAP showed better results in shoot multiplication. The best response was attained on MS medium comprising BAP 5.0 µM + IAA 1.5 µM + ADS 15 µM, with a 20.45 ± 0.12 mean number of shoots and a 6.43 ± 0.006 cm mean shoot length, exhibiting 100% response after 4 weeks of culture. Likewise, the ADS prominence was observed in numerous medicinal plants, such as *Kalanchoe blossfeldiana* [56], *Acacia holosericea* [57], *Rheum spiciforme* [58], and *Anethum graveolens* [59]. In addition to its role in shoot elongation and shoot multiplication, ADS helps to prevent shoot tip necrosis when combined with ammonium sulphate [60]. Hyperhydricity has been reported to be influenced by cytokinin concentration and type [61,62], which is consistent with our present findings. Many of the explants in this investigation that were treated with BAP 5.0 µM + IAA 1.5 µM in addition to varying amounts of ADS and Glu caused mild hyperhydricity in the cultures when they were cultured in flasks (Figure 1e). The shoots did, however, regain their typical appearance after being cultured on a half-strength MS medium that contained auxins for root induction (Figure 2b).

### 2.2. In Vitro Rooting and Acclimatization

Both endogenous and exogenous auxins control the *in vitro* rooting in plant tissue culture [63]. It is generally demonstrated that auxins stimulate the production of ethylene, which inhibits the bud break to cease shooting and promotes the formation of roots from the shoots [64]. *In vitro* rooting was achieved in NAA, IBA, and IAA containing half-strength MS media. High rooting efficiency on half-strength MS medium may likely be due to the reduced macronutrient levels, which are more suitable for root induction compared to shoot development [65]. Among the different auxins tested, IBA (1.5 µM) efficiently produced the highest regeneration, with a 16.44 ± 0.015 mean number of roots per shoot and a mean root length of 2.25 ± 0.011 cm after 6 weeks of culture (Table 4). In the *in vitro* rooting of *A. bracteosa*, 100% success was recorded in some of the literature in a medium containing 0.5 mg L^−1^ IBA [14]. Similarly, ½ MS + IBA was also effective for very high root regeneration in *Juglans mandshurica* [66]. The promotive effect of IBA on root formation has been well-documented across various plant species, with root initiation and development shown to be highly dependent on the presence of exogenous auxins [67]. Several studies within the Lamiaceae family have demonstrated that IBA, either alone at varying concentrations or in combination with other auxins (NAA and IAA), is particularly effective in enhancing rooting parameters, including rooting frequency, root number, and root length [68,69,70,71]. The first root primordia were initiated on half-strength MS medium augmented with IAA 1.5 µM after 2 weeks of culture. Apart from IBA, IAA (1.5 µM) also effectively produced a 15.87 ± 0.020 mean number of roots with a 3.74 ± 0.014 cm mean root length. The roots were short and thick when grown in ½ MS + IBA 0.5 µM, while long and thin roots were obtained in MS + IAA 1.5 µM. In *Salvia sclarea*, IBA induced stronger and more abundant root systems compared to IAA [72]. However, Arikat et al. reported that 2.7 µM IBA not only promoted the longest roots and shoots but also suppressed callus formation [73]. Elmongy et al. also observed similar results with IBA when they screened different auxins for the *in vitro* induction of roots in two *Azalea* cultivars [74]. Conversely, no root induction was observed on IBA 0.5 mgL^−1^ in other species of *Ajuga*, for example, *Ajuga lupulina* [30]. Bhat et al. also obtained the highest root number and mean root length in *Atropa acuminata* in MS medium fortified with 2.0 µM IBA [75]. The superior performance of IBA in rooting has been largely attributed to its enhanced chemical stability during autoclaving and under ambient conditions, relative to other commonly used auxins [76]. Among the different planting substances tested, Soilrite was found to be highly effective for the growth and survival of the plant, with a 90% survival rate (Figure 2d). Among the different planting substances, Soilrite was also found to be the best for the acclimation of *Hemidesmus indicus* [77]. The high survival rate is owing to the high water-holding capacity and highly porous nature of Soilrite. It was seen that during this acclimation phase, the leaves were expanding, and the shoots were growing longer. After successful acclimatization for 4 weeks, plants were transferred into garden soil, showing an 82% survival rate.

Although the present study demonstrates the successful regeneration and acclimatization of *A. bracteosa* plantlets, quantitative estimates of cumulative plant yields across multiple subcultures were not included. To fully validate the efficiency and scalability of this protocol for mass propagation, future work should focus on calculating shoot multiplication rates, rooting efficiency, and survival percentages over successive cycles. Such data will be essential for assessing the practical application of this protocol in commercial or conservation settings.

### 2.3. Genetic Fidelity

Among the several markers, the reproducibility of SCoT markers is higher than that of RAPD and ISSR, and primer length and annealing temperature may not be the only factors influencing reproducibility [78]. *In vitro*-raised plants and the mother plant gave a clear, scorable, distinct, monomorphic band. In the current study, 10 SCoT primers of 18 base pair lengths were employed, which produced monomorphic bands. The monomorphic bands in the SCoT show that the *in vitro*-grown plants were the same in genetic makeup as that of the mother plant (Figure 3). The maximum number of bands was obtained on SCoT 26, producing 11 bands, whereas the least number of bands was obtained on SCoT 14 with 5 bands (Table 5). The results of this study demonstrate that *in vitro* regenerated *A. bracteosa* plantlets retain their genetic homogeneity. Compared to adventitious shoot organogenesis (indirect organogenesis), direct shoot organogenesis is generally associated with greater genetic stability, exhibiting minimal or no genotypic and phenotypic variation [79]. SCoT primers were also employed to evaluate the genetic fidelity of numerous medicinal plants such as *Celatrus paniculatus* and *Cicer arietinum* [80,81]. This is the first publication outlining the use of SCoT primers to assess the genetic fidelity of *A. bracteosa* that have undergone regeneration.

### 2.4. SEM Analysis

Plants grown in a controlled environment showed unusual leaf stomata, exhibiting underdeveloped stomatal complexes lacking bulbous guard cells and open stomatal pores, which may be due to low light conditions, high nutrient availability, and gaseous exchange in the culture tubes. Investigation of the texture and consistency of *in vitro*-derived and acclimatized plants was performed through SEM analysis. *In vitro*-derived leaves had unhealthy, irregular stomata without well-defined guard cells and elongated epidermal cells. Some of the stomata were closed, but most of them were open (Figure 4a). After the transfer of fully acclimatized plants to the greenhouse, a remarkable change in the plants was observed. In the leaves of acclimatized plants, the stomata were well developed and fully functional with normal guard cells, showing normal growth of the plant (Figure 4d). A higher incidence of stomata occurrence is positively correlated with normal leaf development [82]. The photograph also shows that the trichomes of *in vitro*-derived plants were thinner than the trichomes of acclimated plants, which shows the healthy acclimatization behaviour of the plants in the greenhouse conditions (Figure 4b,d). A similar type of study was also performed on plants such as *C. paniculatus* [80] and *H. populifolia* [83].

### 2.5. GC-MS Analysis

Phytochemicals are increasingly recognized not only for their nutritional value but also for their therapeutic and cosmetic applications, particularly due to their ability to neutralize free radicals and mitigate oxidative damage to biological tissues. In this study, GC-MS profiling reveals a total of 32 bioactive compounds in micropropagated *Ajuga bracteosa* plants and 29 in mother plants, underscoring the biochemical competence of *in vitro* cultures as a viable alternative source of medicinally important metabolites. Notably, 13-Docosenamide, (Z)- emerged as the dominant compound in both plant types, suggesting its consistent biosynthetic presence irrespective of growth conditions. This amide has been previously reported for its anti-inflammatory, antimicrobial, and neuroprotective properties, highlighting the potential of *A. bracteosa* extracts in pharmaceutical formulations. Most of the compounds were present in low quantities compared to 13-Docosenamide, (Z)-, which was observed as the main phytocompound present in very high amounts. Among the several secondary metabolites, neophytadiene, erucyl amide, oleic acid amide, β-stigmasterol, erucyl amide, etc., were obtained in both *in vitro* regenerated plants, as well as mother plants (Table 6 and Table 7) (Figure 5 and Figure 6). The presence of these secondary metabolites in micropropagated plants suggests that *in vitro* propagation, particularly under controlled stress and phytohormone-enriched environments, may stimulate or sustain the biosynthetic pathways responsible for secondary metabolite production. This aligns with previous findings that report the stress-induced accumulation of specific phytochemicals under tissue culture conditions [84]. These compounds are more prevalent in methanol, which might be the result of polarity differences that help to solubilize the different compounds. Owing to its strong polarity, methanol has been found to extract several polar phytocompounds (flavonoids and phenolics) from *A. bracteosa* leaves with greater efficiency. Similarly, GC-MS studies were also performed by several researchers using methanol as an extraction solvent in different plant species, such as *Decalepis arayapathra* [85], *Plectranthus amboinicus* [86], *Centratherum punctatum* [87], and *Curcuma zedoaria* [88]. While the presence of key phytocompounds in the regenerants was confirmed through qualitative analysis, the study did not include quantitative comparisons with the mother plant. Such data are essential for evaluating the commercial potential of tissue-cultured plants in secondary metabolite production. Future studies should focus on precise quantification using advanced ultrasensitive techniques, such as HPLC, UPLC-MS, UV/Vis-CM, or UHPLC-Q, to assess the consistency and yield of bioactive compounds in regenerants [89,90]. This will provide deeper insight into the metabolic fidelity of *in vitro*-derived plants and help optimize protocols for large-scale production.

### 2.6. Physiological Studies and Gas-Exchange Analysis

For *in vitro*-regenerated plants, the acclimatization phase is a crucial shift that necessitates significant physiological reprogramming in order to adjust to external environmental circumstances. This study evaluated the photosynthetic recovery and stress adaptability of regenerated *Ajuga bracteosa* plants by monitoring pigment dynamics and gas-exchange parameters. A decrease in chlorophyll a, b, and carotenoid concentration was noted during the first week following *ex vitro* transfer, which could be a sign of photoinhibition or chloroplast dysfunction brought on by the abrupt exposure to intense light [91]. Reduced levels of chl a (0.544 mg gram^−1^), chl b (0.131 mg gram^−1^), and carotenoid (0.256 mg gram^−1^) were all evident during the first week (Figure 7a–c).

The photosynthetic efficiency of the plants increased due to an increase in chlorophyll a (1.274 mg g^−1^), chlorophyll b (0.533 mg g^−1^), and carotenoid content (1.309 mg g^−1^) after 4 weeks of culture, suggesting successful chloroplast remodeling, enhanced light harvesting capacity, and improved photoprotection under *ex vitro* conditions. This progressive recovery was paralleled by a significant enhancement in the net photosynthetic rate (PN) from an initial 0.538 µmol CO_2_ m^−2^ s^−1^, supporting the hypothesis that new leaf development and improved functional stomata contribute to increased photosynthetic efficiency. The same result was also obtained in some medicinal plants, viz. *Celastrus paniculatus* [80].

Underdeveloped cuticles and poorly controlled stomatal activity are the main causes of desiccation, which is the main problem regenerated plantlets encounter during this phase [92]. In order to minimize water loss before stomatal control mechanisms mature, our findings demonstrate that stomatal conductance (Gs) and transpiration rate (E) first fall (Gs: 0.04 to 0.0272 mol m^−2^ s^−1^; E: 0.868 to 0.484 mmol m^−2^ s^−1^). The beginning of functional stomatal development and cuticular thickening, two essential characteristics linked to survival in *ex vitro* settings, is seen in this brief decrease, followed by a progressive recovery. The steady rise in Gs and E over the course of four weeks demonstrates the plant’s capacity to restore gas-exchange regulation and water balance, both of which are essential for transpiration and photosynthesis. Furthermore, this is consistent with earlier acclimatization research on other tissue-cultured plants [93,94], demonstrating that environmental cues, such as a decrease in air vapor pressure and an increase in photon flux density, serve as catalysts for the functional reactivation of stomata. Crucially, these findings imply that gas-exchange metrics are trustworthy markers of plantlet vigor and acclimatization success. The idea that micropropagated *A. bracteosa* has great physiological plasticity, making it appropriate for effective transfer to field conditions, is supported by the consistent improvement across all evaluated features. In conclusion, the research offers useful physiological benchmarks that might be utilized to improve hardening procedures, in addition to showcasing the adaptability of *in vitro*-regenerated *A. bracteosa*. In order to better understand the regulatory networks governing this shift and develop more effective micropropagation techniques, a more comprehensive study combining molecular indicators of stomatal function and photosystem integrity may be necessary.

## 3. Materials and Methods

### 3.1. Plant Material, Media Composition, and Culture Conditions

Young shoots of *Ajuga bracteosa* were collected from an 8-month-old plant that was initially grown in Meelyal (village), Kupwara (tahsil), Jammu, and Kashmir, India, and then transported and cultivated under controlled conditions in a greenhouse. The temperature of the greenhouse was maintained between 15 and 25 °C with a relative humidity of 40–80%, and the plant was watered regularly with tap water. The excised shoots (~5–6 cm) with attached leaves were rinsed immediately after collection, with running tap water for 25 min to eliminate specks of dust and dirt flecks. Afterward, the shoot explants were kept on 1% (*m*/*v*) fungicide Bavistin (BASF, Mumbai, India) for 25 min and then rinsed thoroughly under tap water. Thereafter, 5% (*v*/*v*) liquid detergent Teepol was used to wash the explants, and then, they were washed 6–7 times with sterilized double-distilled water (DDW). Subsequently, the explants were kept in 0.1% (*m*/*v*) mercuric chloride solution for 3–4 min under a laminar flow hood, followed by washing 5–6 times with autoclaved DDW. From the collected shoots, internodal segments of ~1.0 cm were used as explants for the inoculation.

To prepare Murashige and Skoog (MS) [24] stock solutions, concentrated solutions of macronutrients (20×), micronutrients (200×), vitamins (100×), and iron (100×) were made separately by dissolving each group of components in autoclaved double-distilled water. The chemicals were first dissolved in a small volume of water with gentle stirring and mild heating, and then the solution was brought up to the final volume using a volumetric flask. All prepared stock solutions were clearly labeled and stored at 4 °C until used for the preparation of MS medium.

MS medium with different plant growth regulators (PGRs), containing 3% (*m*/*v*) sucrose (Qualigens, India) and 0.8% (*m*/*v*) agar (Qualigens, India), was used during the experiment. The adjustment of pH to 5.8 was performed by adding 1 N NaOH and 1 N HCl. After adjusting the pH, agar was dissolved using a microwave, and a clear solution was poured into the culture vials (25 × 150 mm, Borosil, India) and a wide-mouth flask (100 mL, Borosil, India) and then plugged with non-absorbent cotton wrapped in muslin cloth. Hormone-free MS medium was treated as a control. The culture media, DDW, and other utensils were sterilized by autoclaving at 15 lb pressure, generating a temperature of 121 °C, for 15 min. After the inoculation of internodal explants, the culture vials were incubated at 55 ± 5% relative humidity, 25 ± 2 °C temperature, and a 16/8 h photoperiod of 50 µmol m^−2^ s^−1^ of photosynthetic photon flux density (PPFD) equipped with fluorescent tubes (40 W; Philips, India) throughout the experiment.

### 3.2. Shoot Induction and Multiplication

For the regeneration of the axillary shoot, internodal sections without leaves were used. Internodal sections of 1.0 cm were inoculated aseptically on MS medium augmented with different concentrations of cytokinins, such as 6-benzyl-amino-purine (BAP), kinetin (Kn), and 2-isopentanyladenine (2iP), at concentrations of 1.5, 2.5, 5.0, and 7.5 µM. To intensify the regeneration efficiency, the optimized cytokinin (BAP 5.0 µM) that showed maximum shoot proliferation was augmented with different auxins (α-naphthalene acetic acid (NAA), indole-3-acetic acid (IAA), and indole-3-butyric acid (IBA)) in various strengths (0.5, 1.0, 1.5, and 2.0 µM). Optimized combination treatments of cytokinin and auxin, along with adenine sulphate (ADS) and glutamine (GLU) (5.0, 10.0, 15.0, and 20.0 µM), were used to evaluate further enhancement in the multiplication and proliferation of shoots. The culture vials were maintained under controlled conditions of 55 ± 5% relative humidity, a temperature of 25 ± 2 °C, and a 16 h light/8 h dark photoperiod, with a photosynthetic photon flux density (PPFD) of 50 µmol m^−2^ s^−1^ provided by 40 W fluorescent tubes (Philips, India) throughout the experiment, with each phase lasting 4 weeks.

### 3.3. Rooting, Acclimatization, and Field Relocation

For *in vitro* rooting, the axial microshoots from 28-day-old cultures, measuring 4–5 cm in length, were transferred onto half-strength MS medium with or without auxins (IBA, IAA, and NAA) at altered concentrations (0.5, 1.5, and 2.5 µM). MS medium without auxins was used as a control. The rooted plantlets, after the full development of roots and six weeks of emergence, were removed from the culture vials, washed softly to eliminate the traces of media, and shifted to Polyethylene Pots that contained autoclaved garden soil, vermicompost, and Soilrite^TM^ (Irish peat and horticulture grade expanded perlite in 3:1) (Keltech Energies, Bangalore, India). The plants were covered with thin polythene for 3 weeks and kept in a culture room under 55 ± 5% relative humidity, a temperature of 25 ± 2 °C, and a 16/8 h photoperiod. To gradually acclimate the plants to their surroundings, the polythene cover was opened and closed for 12 h each, over two weeks in the culture room. Moreover, for 1 week, the plantlets were moistened with DDW water containing MS I [24], and thereafter, the plants were watered with single-distilled water followed by tap water. After successful acclimation, the plantlets were allocated in clay pots comprising a 2:1 ratio of garden soil to green manure, and they were housed in the culture room for six weeks, following another four weeks in the greenhouse. Thereafter, they were moved to the field where they were cultivated in full sunlight.

### 3.4. Genetic Fidelity Assessment Using SCoT Marker

For the determination of genetic fidelity, ten fully acclimatized 8-week-old plantlets of *A. bracteosa* were randomly chosen to compare with the 16-month-old mother plant that was grown in the greenhouse. The temperature of the greenhouse was maintained between 15 and 25 °C, with a relative humidity of 40–80%, and the plant was watered regularly with tap water. Subsequently, genomic Deoxyribonucleic acid (DNA) was extracted from ~10–15 mg of lyophilized leaf tissue using a modified Cetyltrimethylammonium Bromide (CTAB) method [95]. The tissue was ground and mixed with pre-warmed CTAB buffer containing 0.2% β-mercaptoethanol and then incubated at 65 °C for 1 h. After cooling, chloroform/isoamyl alcohol (24:1) was added for phase separation. The aqueous layer was collected, treated with Ribonuclease A (10 mg/mL), and re-extracted to remove contaminants. DNA was precipitated using cold isopropanol (0.6 volumes), pelleted by centrifugation at 5000 rpm for 3 min at 4 °C, washed with ethanol containing ammonium acetate (10 mM ammonium acetate in 70% ethanol), air-dried, and dissolved in nuclease-free water. The purity was checked using a Ultraviolet-Visible (UV-Vis) spectrophotometer at 260/280 nm wavelength. In the present study, a set of 10 SCoT primers was used to carry out polymerase chain reaction (PCR) in a thermocycler (Biometra T Gradient Thermoblock, Analytik Jena, Göttingen, Germany). The procedures of Rohela et al. were used for the separation of the reaction mixture and amplicons during PCR amplification [96]. The PCR reaction was performed in a 10 µL volume containing 25 ng of template DNA, 10 x Taq buffer, 10 mM deoxyribonucleoside triphosphate, 25 mM MgCl_2_, 5 u/µL Taq DNA polymerase, and 10 mM primer. The cyclic conditions for SCoT analysis included denaturation for 5 min at 94 °C, followed by 35 cycles of 30 s at 94 °C, the annealing temperature was 50 °C for 30 s, the extension temperature was 72 °C for 1 min, and the final extension was for 5 min at 72 °C.

### 3.5. Observation of Stomatal Structures via SEM

*In vitro*-grown plants from culture vials, as well as four-week-old acclimatized plants grown in the greenhouse (15–25 °C temperature and 40–80% relative humidity), were employed for scanning electron microscopy (SEM) analysis. Leaves were collected and fixed in 3% glutaraldehyde for two hours, which was prepared in phosphate buffer. Thereafter, the leaves were transferred to 30% ethyl alcohol for 15 min to initiate the dehydration. Subsequently, the leaves were transferred to a 60% alcohol solution for another 15 min, followed by immersion in 90% alcohol for an additional 15 min. Finally, the leaves were placed in absolute (100%) alcohol for a 15 min duration for further dehydration. After dehydration, the samples were critical point dried, and both the surfaces of leaves were plated with gold (Gold Sputter Coater Unit, JFC-1600, JEOL, Tokyo, Japan) and then mounted on aluminium stubs. Analysis was performed on a SEM (JSM-6510, LV-JEOL, Tokyo, Japan) with an accelerating voltage of 15 kV. The images of the SEM were presented digitally on a computer screen.

### 3.6. Preparation of Plant Methanolic Extract for GC-MS Analysis and Identification of Chemical Constituents

Fully developed leaves of four-week-old, acclimatized plants and the mother plant were selected for GC-MS investigation. The leaves were gathered, thoroughly washed, and then dried in an oven at 60 °C for 24 h. Once dried, they were ground using a mortar and pestle until a fine powder was obtained. One gram of this powdered sample was mixed with 50 mL of methanol and left to stand for 24 h for extraction. The resulting mixture was then centrifuged at 5000 RCF for 5 min. To further purify the extract, the supernatant was filtered through a 0.22 µm syringe filter (Genetix, India).

GC-MS analysis was carried out using Column-Rxi-5MS (30 m × 0.25 i.d. × 0.25 μm film thickness) and GC-MS QP-2010 Ultra (Shimadzu Company, Kyoto, Japan). Helium (99.999%) was utilized as the carrier gas in split mode (10:1) at a flow rate of 1 milliliter per minute. A volume of approximately 8 μL was injected into a column at 260 °C for the injector and 270 °C for the detector. The oven’s temperature began at 80 °C and stayed there for two minutes before increasing by 10 °C every minute until 250 °C without holding. The ion source’s temperature was kept at 230 °C, while holding was permitted for 10 min at 280 °C at a program rate of 5 °C per minute. The detector ran in scan mode from 45 to 450 Da atomic mass units, and the mass spectrum of the compounds contained in the extracts was obtained by electron ionization at 70 eV. Fragments ranging from 45 to 450 Da and a scan interval of 0.5 s were preserved. It ran for 55 min in total. The structure of the molecules, masses, and fragments determined the compound’s identity. The National Institute of Standards and Technology (NIST) database, which contains over 62,000 patterns, was used to elucidate mass spectra GC-MS. The components of the test materials were identified by their IUPAC names, molecular weights, and structures. Using information stored in the NIST library version (2005) and the software Turbomas 5.2, a comparative analysis of the spectra of known and unknown substances was conducted.

### 3.7. Physiological Analysis

Fully expanded leaf samples of *in vitro*-developed rooted plantlets were used to perform physiological and biochemical studies. Twelve plantlets, which were kept in the culture room, were randomly selected and investigated at 0 days, 1 week, 2 weeks, and 3 weeks, as well as at 4 weeks of the acclimatization period.

#### 3.7.1. Estimation of Chlorophyll and Carotenoid Content

Chlorophyll and carotenoid contents were assessed using the technique applied by Ahmad et al. [97]. An 80% (*v*/*v*) acetone solution was used to grind the samples in a mortar and pestle for the extraction of pigments. The samples were purified with Whatman’s No. 1 filter paper. The optical density (O.D.) values of chlorophyll a, chlorophyll b, and carotenoid contents were measured at wavelengths of 662, 645, and 470 nm, respectively, in an Ultraviolet-Visible (UV-Vis) spectrophotometer (UV-1700 Pharma Spec, Shimadzu, Kyoto, Japan).

#### 3.7.2. Net Photosynthetic Rate, Stomatal Conductance, and Transpiration Rate

For the estimation of PN (net photosynthetic rate), Gs (stomatal conductance), and E (transpiration rate), the fully expanded upper leaf of the main axis of the plant was selected. An infrared gas analyzer (IRGA) (Li-COR 6400, LI-COR, and Lincoln, NE, USA) was employed at 800 μmol/m^2^ s photosynthetically active radiation (PAR) from 12:00 a.m. to 1:00 p.m. The experiments were conducted on *in vitro*-grown plants over several weeks of acclimatization. These parameters were computed using the techniques described by Yadav et al. [98].

### 3.8. Statistical Analysis

For statistical analysis, 20 replicates per treatment were conducted, and all the experiments were repeated three times. The following rates were calculated for each replicate: shoot-induction rate (%) = (number of explants producing ≥1 shoot/total explants) × 100; rooting rate (%) = (number of shoots forming roots/total shoots transferred to rooting medium) × 100; and acclimatization (survival) rate (%) = (number of plantlets surviving after hardening/total plantlets transferred *ex vitro*) × 100. Morphological observations, such as shoot and root number, as well as shoot and root length, were recorded regularly. One-way ANOVA was used in all statistical analyses of the data using SPSS version. 25 (SPSS Inc., Chicago, IL, USA). Duncan’s multiple range test (DMRT) was used to compare significant differences between means at *p* ≤ 0.05. Mean ± standard error (SE) was used to express the data’s variability.

## 4. Conclusions

A significant amount of high-quality pharmaceutical goods could be produced via *in vitro* propagation. A successful approach to propagating *A. bracteosa* can be achieved through an improved and reliable *in vitro* method using internodal explants. The shoot was initiated on BA (5.0 µM), and the best response for the multiplication of shoots was achieved on BA (5.0 µM) + IAA (1.5 µM) + ADS (15 µM), while for rooting, the best response was observed on half-strength MS media augmented with IBA (1.5 µM).

The best response for the multiplication of shoots was achieved on BA (5.0 µM) + IAA (1.5 µM) + ADS (15 µM), while for rooting, the best response was observed on half-strength MS medium augmented with IBA (1.5 µM). The genetic stability and true-to-type characteristics of the micropropagated plants were confirmed by the DNA-based markers. DNA-based SCoT marker analysis demonstrates monomorphic banding patterns between regenerants and the mother plant; however, we acknowledge that more comprehensive molecular and cytogenetic studies (e.g., additional marker systems and flow cytometry) are required to conclusively rule out somaclonal variation. A suite of physiological assessments, chlorophyll a and b, carotenoid content, and stomatal conductance confirmed the successful acclimatization of plantlets, and GC-MS profiling verified the preservation of key phytoconstituents. Altogether, this protocol lays a robust foundation for selecting superior genotypes, facilitating genetic transformation, and supporting both conservation and commercial applications.

## Figures and Tables

**Figure 1 plants-14-02507-f001:**
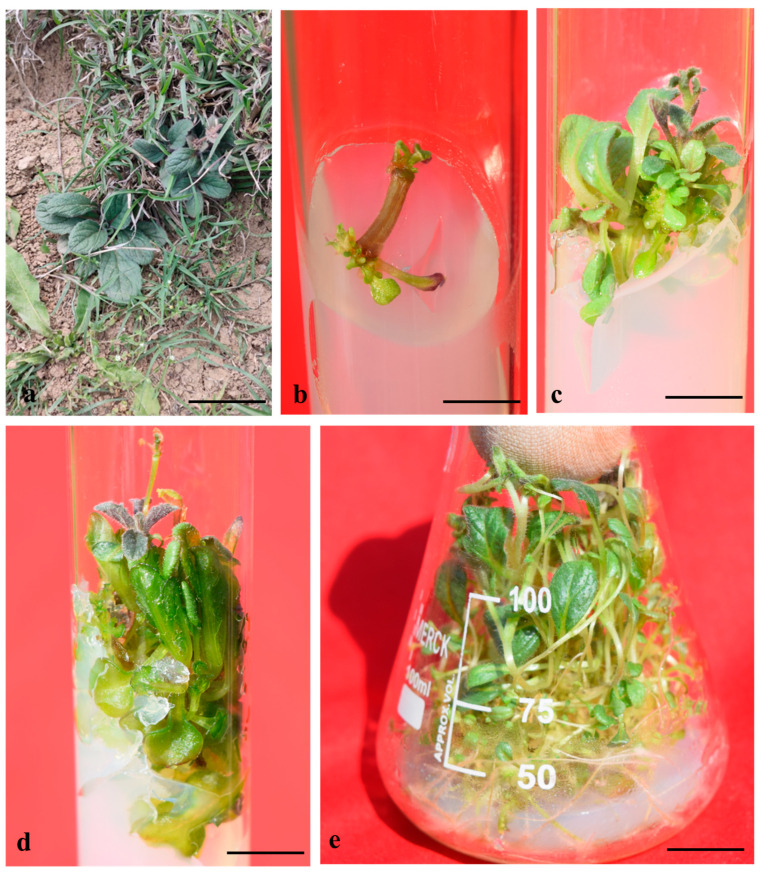
Establishment of *A. bracteosa* culture through internodes. Mother plant (**a**), internodal segment at MS + BA 5 µM—1-week-old culture (bar: 1 cm = 0.6 cm) (**b**), shoot regeneration on MS + BA 5 µM—4-weeks-old culture (bar: 1 cm = 0.96 cm) (**c**), shoot elongation on MS + BA 5 µM + IAA 1.5 µM—4-weeks-old culture (bar: 1 cm = 1.13 cm) (**d**), multiplication and proliferation of shoots on MS + BA 5 µM + IAA 1.5 + ADS 15 µM—4-weeks-old culture (bar: 1 cm = 1.38 cm) (**e**).

**Figure 2 plants-14-02507-f002:**
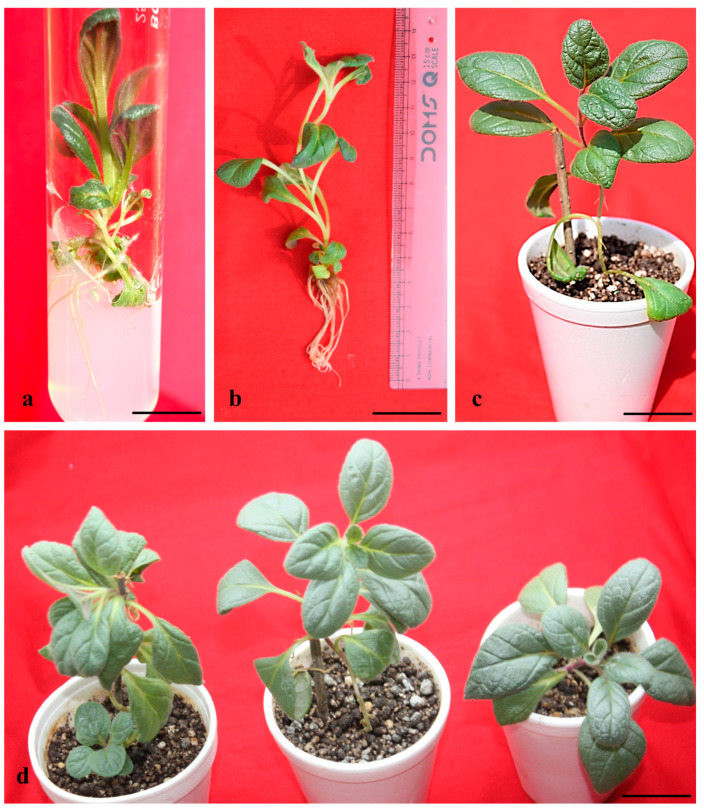
*In vitro* rooting on MS ½ medium + IAA 1.5 µM—6-week-old culture (bar: 1 cm = 1.56 cm) (**a**), *in vitro*-rooted microshoots—6 weeks of culture (bar: 1 cm = 2.57 cm) (**b**), acclimatized plantlets in Soilrite—4 weeks old (bar: 1 cm = 8.81 cm) (**c**), (bar: 1 cm = 9.85 cm) (**d**).

**Figure 3 plants-14-02507-f003:**
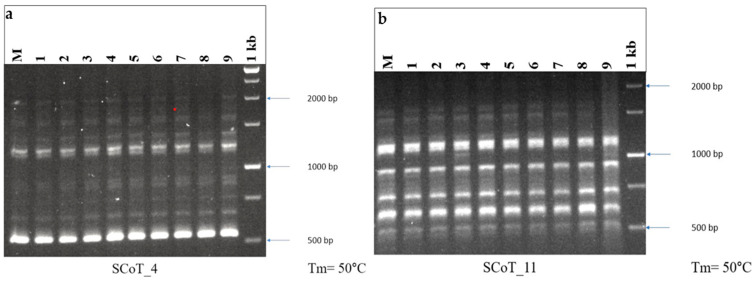
DNA fingerprinting profile of micropropagated plants and mother plant of *A. bracteosa* through (**a**) SCoT_4 and (**b**) SCoT_11, showing monomorphic banding pattern. M = mother plant, Lane 1–9 = *in vitro*-regenerated randomly selected plants, 1 kb = DNA ladder.

**Figure 4 plants-14-02507-f004:**
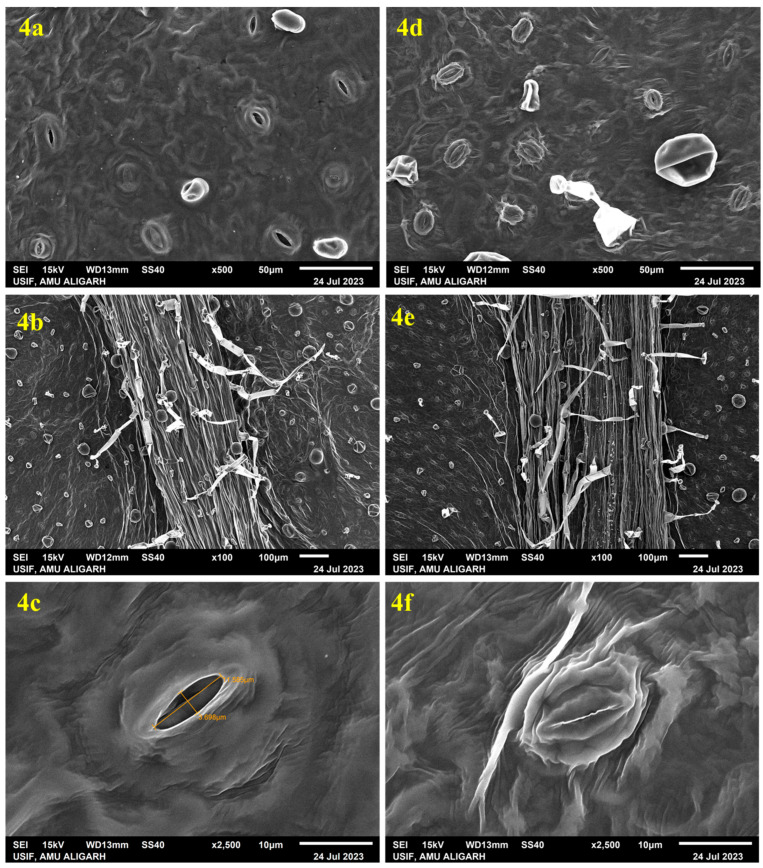
Scanning electron microscope (SEM) observation of *in vitro*-derived leaf of *A. bracteosa*: (**4a**) abaxial surface showing wide-open stomata with unhealthy guard cells and irregular stomatal aperture, (**4b**) abaxial surface of leaf showing small-sized trichomes, (**4c**) single wide-open stoma showing ruptured guard cell. SEM observation of a leaf excised from acclimatized *A. bracteosa*: (**4d**) abaxial surface showing numerous well-developed stomata, (**4e**) abaxial surface of leaf showing large-sized trichomes, (**4f**) single closed stoma showing developed guard cell.

**Figure 5 plants-14-02507-f005:**
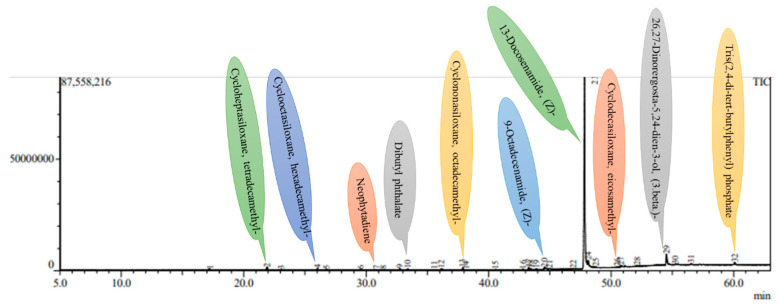
A GC-MS chromatogram of the methanolic leaf extract of the mother plant of *A. bracteosa*.

**Figure 6 plants-14-02507-f006:**
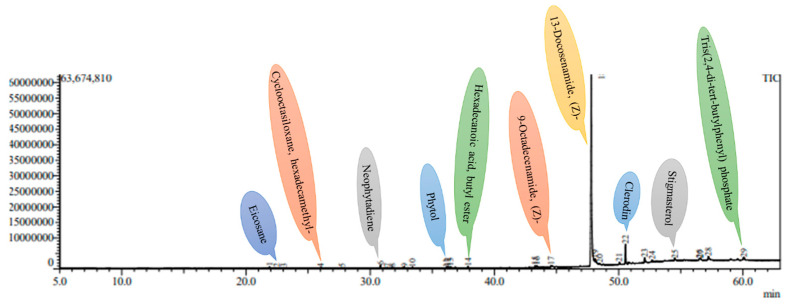
A GC-MS chromatogram of the methanolic leaf extract of a 4-week-old *in vitro*-derived plantlet of *A. bracteosa*.

**Figure 7 plants-14-02507-f007:**
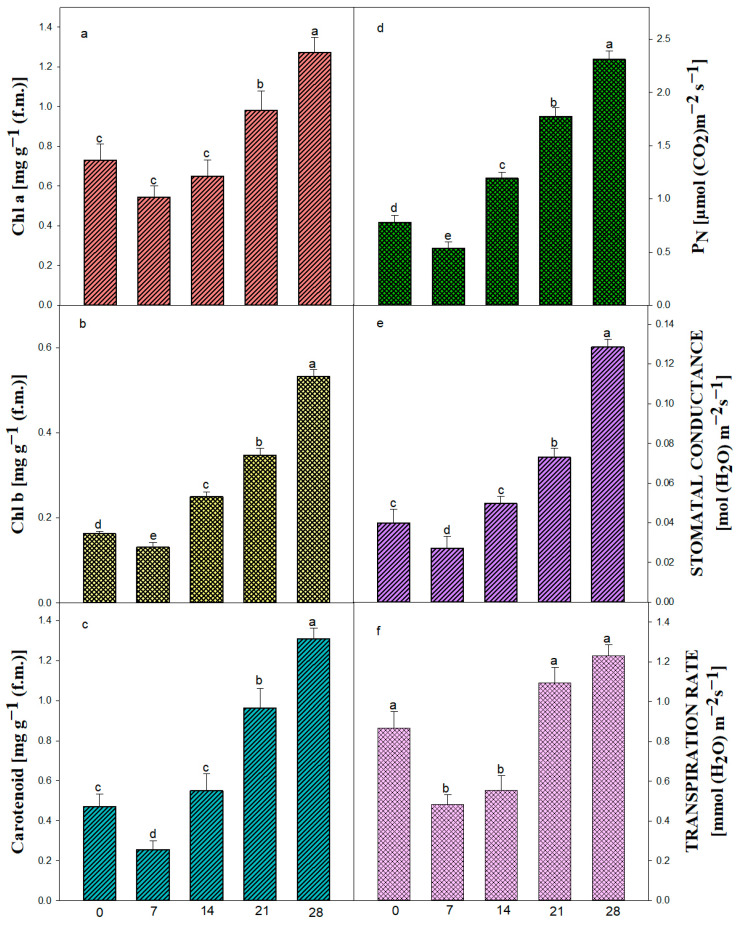
Pigment content and related characteristics in *in vitro*-raised plantlets of *A. bracteosa*. Chlorophyll a (**a**), Chlorophyll b (**b**), carotenoid (**c**), net photosynthetic rate (PN) (**d**), stomatal conductance (Gs) (**e**), transpiration rate (**f**). Bars represent mean ± SE. Bars represented by the same letter within the response variable are not significantly different (*p* ≤ 0.05) according to Tukey’s test at 5% probability.

**Table 1 plants-14-02507-t001:** Effect of different cytokinins on *in vitro* shoot induction of *A. bracteosa* after 4 weeks of culture *.

Cytokinin (µM)	% Response	Mean No. of Shoots	Mean Shoot Length (cm)
Control	-	-	-
BAP 1.5	72 ± 0.57 ^d^	7.66 ± 0.023 ^c^	1.70 ± 0.115 ^e^
BAP 2.5	85 ± 1.15 ^b^	9.14 ± 0.272 ^b^	2.43 ± 0.088 ^c^
BAP 5.0	94 ± 0.57 ^a^	9.98 ± 0.057 ^a^	2.60 ± 0.057 ^a^
BAP 7.5	77 ± 1.73 ^c^	5.49 ± 0.008 ^e^	2.33 ± 0.033 ^b^
2iP 1.5	60 ± 1.15 ^f^	4.76 ± 0.017 ^h^	1.10 ± 0.057 ^h^
2iP 2.5	67 ± 1.15 ^e^	5.44 ± 0.008 ^f^	1.33 ± 0.033 ^g^
2iP 5.0	80 ± 0.57 ^c^	5.89 ± 0.005 ^d^	2.16 ± 0.088 ^d^
2iP 7.5	64 ± 1.73 ^e^	4.71 ± 0.017 ^i^	1.86 ± 0.066 ^e^
Kn 1.5	31 ± 0.57 ^i^	2.26 ± 0.020 ^l^	0.80 ± 0.057 ^i^
Kn 2.5	45 ± 0.57 ^g^	4.44 ± 0.012 ^k^	1.30 ± 0.057 ^f^
Kn 5.0	65 ± 2.30 ^e^	5.15 ± 0.017 ^g^	1.70 ± 0.100 ^e^
Kn 7.5	39 ± 0.57 ^h^	4.64 ± 0.020 ^j^	1.33 ± 0.088 ^g^

* Each experiment was conducted three times with 20 replicates. The values represent the mean  ±  standard error. The same letters within the columns are not significantly different according to Duncan’s New Multiple Range Test at a 5% level (*p* ≤  0.05). BAP, 6-benzyl-amino-purine; 2iP, 2-isopentanyladenine; Kn, kinetin.

**Table 2 plants-14-02507-t002:** Effect of different combinations of auxin with BAP (5.0 µM) on shoot regeneration through the internodal segment of *A. bracteosa* after 4 weeks of culture *.

Auxin (µM)	% Response	Mean No. of Shoots	Mean Shoot Length (cm)
NAA 0.5	83 ± 1.15 ^e^	10.81 ± 0.092 ^d^	3.05 ± 0.014 ^e^
NAA 1.0	92 ± 0.57 ^c^	11.63 ± 0.051 ^c^	3.33 ± 0.035 ^c^
NAA 1.5	100 ± 0.00 ^a^	12.15 ± 0.014 ^a^	3.79 ± 0.017 ^b^
NAA 2.0	90 ± 0.57 ^c^	11.80 ± 0.079 ^b^	3.22 ± 0.012 ^d^
IBA 0.5	87 ± 1.15 ^d^	8.25 ± 0.006 ^i^	2.23 ± 0.057 ^h^
IBA 1.0	90 ± 0.57 ^c^	8.36 ± 0.017 ^h^	2.55 ± 0.030 ^g^
IBA 1.5	100 ± 0.00 ^a^	9.54 ± 0.012 ^g^	2.93 ± 0.017 ^f^
IBA 2.0	96 ± 0.57 ^b^	8.44 ± 0.008 ^h^	2.53 ± 0.013 ^g^
IAA 0.5	85 ± 0.57 ^de^	9.57 ± 0.085 ^g^	3.24 ± 0.034 ^d^
IAA 1.0	100 ± 0.00 ^a^	9.82 ± 0.032 ^f^	3.73 ± 0.021 ^b^
IAA 1.5	100 ± 0.00 ^a^	12.25 ± 0.028 ^a^	4.19 ± 0.035 ^a^
IAA 2.0	87 ± 1.15 ^d^	10.05 ± 0.029 ^e^	3.34 ± 0.008 ^c^

* Each experiment was conducted three times with 20 replicates. The values represent the mean  ±  standard error. The same letters within the columns are not significantly different according to Duncan’s New Multiple Range Test at a 5% level (*p* ≤  0.05). NAA, α-naphthalene acetic acid; IBA, indole-3-butyric acid; IAA, indole-3-acetic acid.

**Table 3 plants-14-02507-t003:** Effect of different growth additives on shoot regeneration from internodal segments of *A. bracteosa* after 4 weeks of culture *.

Additives (µM)	% Response	Mean No. of Shoots	Mean Shoot Length (cm)
ADS 5	76 ± 1.15 ^e^	10.46 ± 0.18 ^e^	5.27 ± 0.008 ^d^
ADS 10	85 ± 0.57 ^d^	13.49 ± 0.20 ^c^	5.63 ± 0.012 ^c^
ADS 15	100 ± 0.00 ^a^	20.45 ± 0.12 ^a^	6.43 ± 0.006 ^a^
ADS 20	95 ± 1.15 ^b^	18.62 ± 0.28 ^b^	5.88 ± 0.017 ^b^
GLU 5	65 ± 0.57 ^f^	6.52 ± 0.17 ^h^	2.55 ± 0.017 ^h^
GLU 10	85 ± 1.73 ^d^	8.44 ± 0.12 ^g^	3.55 ± 0.015 ^g^
GLU 15	95 ± 1.15 ^b^	11.42 ± 0.26 ^d^	4.16 ± 0.017 ^e^
GLU 20	89 ± 0.57 ^c^	9.24 ± 0.017 ^f^	3.96 ± 0.008 ^f^

* Each experiment was conducted three times with 20 replicates. The values represent the mean ±  standard error. The same letters within the columns are not significantly different according to Duncan’s New Multiple Range Test at a 5% level (*p* ≤ 0.05). ADS, adenine sulphate; GLU, Glutamine.

**Table 4 plants-14-02507-t004:** Effect of different auxins on *in vitro* root induction in *A. bracteosa* after 6 weeks of culture *.

½ MS + Auxins (µM)	Rooting %	Mean No. of Roots	Mean Root Length (cm)
Control	-	-	-
IAA 0.5	89 ± 0.57 ^c^	13.35 ± 0.035 ^c^	2.90 ± 0.020 ^b^
IAA 1.5	100 ^a^	15.87 ± 0.020 ^b^	3.74 ± 0.014 ^a^
IAA 2.5	100 ^a^	12.47 ± 0.025 ^d^	1.35 ± 0.017 ^d^
IBA 0.5	82 ± 1.15 ^d^	8.82 ± 0.032 ^h^	1.11 ± 0.012 ^f^
IBA 1.5	100 ^a^	16.44 ± 0.015 ^a^	2.25 ± 0.011 ^c^
IBA 2.5	100 ^a^	11.96 ± 0.020 ^f^	1.24 ± 0.008 ^e^
NAA 0.5	100 ^a^	7.57 ± 0.030 ^i^	0.96 ± 0.008 ^g^
NAA 1.5	100 ^a^	12.35 ± 0.017 ^e^	1.37 ± 0.003 ^d^
NAA 2.5	95 ± 1.73 ^b^	11.57 ± 0.025 ^g^	0.84 ± 0.012 ^h^

* Each experiment was conducted three times with 20 replicates. The values represent the mean  ±  standard error. The same letters within the columns are not significantly different according to Duncan’s New Multiple Range Test at a 5% level (*p* ≤ 0.05). NAA, α-naphthalene acetic acid; IBA, indole-3-butyric acid; IAA, indole-3-acetic acid.

**Table 5 plants-14-02507-t005:** SCoT primers with their corresponding sequences and number of bands in the *A. bracteosa* stock plant.

Sr. No.	Primer	Sequences (5′-3′)	%G/C	Tm	Total Band
1	SCoT 2	CAACAATGGCTACCACCC	56	50 °C	7
2	SCoT 4	CAACAATGGCTACCACCT	50	50 °C	10
3	SCoT 7	CAACAATGGCTACCACGG	56	50 °C	9
4	SCoT 9	CAACAATGGCTACCAGCA	50	50 °C	9
5	SCoT 11	AAGCAATGGCTACCACCA	50	50 °C	8
6	SCoT 14	ACGACATGGCGACCACGC	56	50 °C	5
7	SCoT 17	ACCATGGCTACCACCGAG	67	50 °C	7
8	SCoT 19	ACCATGGCTACCACCGGC	67	50 °C	7
9	SCoT 26	ACCATGGCTACCACCGTC	61	50 °C	11
10	SCoT 29	CCATGGCTACCACCGGCC	72	50 °C	10
Total					83

**Table 6 plants-14-02507-t006:** GC-MS data of the methanolic extract of a micropropagated plant of *A. bracteosa*.

Peak#	Compounds	R. Time	Area	Area%	Height
1.	Cyclohexasiloxane, dodecamethyl-	17.248	4,963,145	0.87	454,300
2.	Cycloheptasiloxane, tetradecamethyl-	21.912	4,083,360	0.71	1,188,873
3.	2,4-Di-tert-butylphenol	23.070	1,075,361	0.19	382,584
4.	Cyclooctasiloxane, hexadecamethyl-	26.032	2,090,936	0.37	810,023
5.	1,4-Cyclohexanediol, (Z)-	26.749	1,172,177	0.20	272,590
6.	Cyclononasiloxane, octadecamethyl-	29.565	821,456	0.14	342,126
7.	Neophytadiene	30.849	753,058	0.13	263,833
8.	1,2-Benzenedicarboxylic acid, bis(2-methylpropyl) ester	31.378	1,032,343	0.18	276,252
9.	Hexadecanoic acid, methyl ester	32.758	1,034,573	0.18	438,958
10.	Dibutyl phthalate	33.396	1,733,556	0.30	541,914
11.	Cyclooctasiloxane, hexadecamethyl-	35.566	503,021	0.09	207,479
12.	9,12-Octadecadienoic acid (Z, Z)-, methyl ester	36.130	1,450,340	0.25	481,297
13.	Hexadecanoic acid, butyl ester	37.885	3,282,875	0.57	1,105,824
14.	Cyclononasiloxane, octadecamethyl-	38.159	527,651	0.09	213,131
15.	Cyclodecasiloxane, eicosamethyl-	40.554	527,917	0.09	210,801
16.	Cyclodecasiloxane, eicosamethyl-	42.812	913,969	0.16	269,439
17.	(9Z)-Docosenenitrile	43.273	2,517,431	0.44	808,941
18.	13-Docosenoic acid, methyl ester	43.437	1,083,194	0.19	372,339
19.	Bis(2-ethylhexyl) phthalate	43.903	1,114,285	0.19	308,229
20.	9-Octadecenamide, (Z)-	44.557	3,113,968	0.54	926,303
21.	Hexadecanamide	44.979	2,698,177	0.47	373,180
22.	Cyclononasiloxane, octadecamethyl-	46.920	1,042,641	0.18	340,392
23.	13-Docosenamide, (Z)-	47.832	485,760,325	84.86	86,682,267
24.	Tetradecanamide	48.131	6,764,006	1.18	1,853,862
25.	Polydimethylsilane Oil	48.764	1,169,324	0.20	375,342
26.	Cyclodecasiloxane, eicosamethyl-	50.508	2,191,935	0.38	378,105
27.	13-Docosenamide, (Z)-	50.810	4,121,176	0.72	709,706
28.	Cyclodecasiloxane, eicosamethyl-	52.136	1,552,021	0.27	422,022
29.	26,27-Dinorergosta-5,24-dien-3-ol, (3 beta)-	54.524	25,070,824	0.38	4,606,996
30.	β-Stigmasterol	55.243	1,448,270	0.25	245,084
31.	Silane, diethylheptyloxyoctadecyloxy-	56.523	1,216,167	0.21	337,767
32.	Bracteatin	60.099	5,576,346	0.97	987,945

**Table 7 plants-14-02507-t007:** GC-MS data of the methanolic extract of the mother plant of *A. bracteosa*.

Peak#	Compounds	R. Time	Area	Area%	Height
1.	Cycloheptasiloxane, tetradecamethyl-	21.906	1,236,170	0.31	308,863
2.	Eicosane	22.485	631,453	0.16	170,749
3.	2,4-Di-tert-butylphenol	23.063	742,530	0.18	206,353
4.	Cyclooctasiloxane, hexadecamethyl-	26.020	882,777	0.22	274,405
5.	Eicosane	27.730	698,563	0.17	216,933
6.	Neophytadiene	30.839	2,448,887	0.61	905,081
7.	Phthalic acid, 2-cyclohexylethyl isobutyl ester	31.372	917,185	0.23	287,096
8.	3,7,11,15-Tetramethyl-2-hexadecen-1-ol	31.769	1,118,705	0.28	337,869
9.	Hexadecanoic acid, methyl ester	32.759	1,009,634	0.25	263,188
10.	Dibutyl phthalate	33.403	1,384,013	0.34	307,349
11.	9,12-Octadecadienoic acid (Z, Z)-, methyl ester	36.131	848,285	0.21	272,069
12.	Dichloroacetic acid, tridec-2-ynyl ester	36.257	1,658,903	0.41	432,989
13.	Phytol	36.463	1,634,769	0.41	431,251
14.	Hexadecanoic acid, butyl ester	37.883	1,724,784	0.43	590,789
15.	9-Octadecenenitrile, (Z)-	43.267	1,579,434	0.39	494,260
16.	13-Docosenoic acid, methyl ester	43.425	1,382,227	0.34	473,881
17.	9-Octadecenamide, (Z)-	44.560	2,540,082	0.63	609,542
18.	13-Docosenamide, (Z)-	47.800	318,983,333	79.47	62,068,055
19.	9-Octadecenamide, (Z)-	48.122	4,057,362	1.01	1,040,441
20.	Tricyclo[5.3.1.1(2,6)]dodecan-11-ol, 11-methyl-12-methylene	48.477	891,996	0.22	309,582
21.	.beta.-Phenoxyethyl methacrylate	50.059	2,168,888	0.54	676,135
22.	Clerodin	50.557	23,257,784	5.79	6,320,294
23.	.beta.-Phenoxyethyl methacrylate	52.108	5,994,264	1.49	1,547,154
24.	6-Hydroxyeurycomalactone	52.697	2,747,853	0.68	554,415
25.	Stigmasterol	54.523	3,793,498	0.95	710,354
26.	Phenol, 2,4-bis(1,1-dimethylethyl)-, phosphite (3:1)	56.507	1,968,100	0.49	431,511
27.	Ursa-12,20(30)-dien-28-oic acid, 2,3,23-trihydroxy-, (2.alpha.,3.beta.,4.alpha.)-, 4Me derivative	56.591	2,713,988	0.68	615,422
28.	Micromeric acid methyl ester	57.218	7,351,153	1.83	1,078,460
29.	Tris(2,4-di-tert-butylphenyl) phosphate	60.078	5,029,600	1.25	877,551

## Data Availability

The original contributions presented in this study are included in the article. Further inquiries can be directed to the corresponding author(s).

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
