# Peer review of "Micropropagation of Ajuga bracteosa via Direct Organogenesis Using Internodal Explants: SEM, GC-MS, and SCoT Marker Analysis"

_plants, 2025, doi:10.3390/plants14162507_

Round 1

Reviewer 1 Report (Previous Reviewer 2)

Comments and Suggestions for Authors

The article is interesting and contributed to the development of tissue culture in this species. Ajuga bracteosa is an herb with high medicinal value and limited distribution. This article suggests that the in vitro protocol for Ajuga bracteosa can significantly facilitate genetic transformation, supporting conservation and commercial applications.

The authors have significantly improved the article. 

Author Response

We sincerely thank you for your positive and encouraging feedback. We are grateful for your recognition of the significance of our work on Ajuga bracteosa and its potential contributions to conservation, commercial application, and genetic transformation. Your comments have greatly motivated us, and we appreciate your thoughtful review and support in improving the quality of the manuscript.

Reviewer 2 Report (Previous Reviewer 3)

Comments and Suggestions for Authors

Thanks, good luck

Author Response

We sincerely thank you for your positive and encouraging feedback. Your comments have greatly motivated us, and we appreciate your thoughtful review and support in improving the quality of the manuscript.

Reviewer 3 Report (New Reviewer)

Comments and Suggestions for Authors

The study focuses on the in vitro propagation of Ajuga bracteosa using internodal explants. Although micropropagation in this species is not novel, the manuscript is interesting and presents a solid dataset that contributes to the existing body of knowledge. However, some aspects of the Materials and Methods section require clarification. Grammatical review is needed to ensure consistency along the document. The reference list should also be revised to conform to the journal’s formatting guidelines. A detailed review is provided below.

Line 12: Revise to - Correspondence: ashahzad.bt@amu.ac.in (A.S.); ahmad.lycos@gmail.com or ahmad.bot@njfu.edu.cn (Z.A.)

Lines 20 and 23: “media” or “medium”. Media is plural and medium is singular – revise accordingly – revise this throughout the document

Line 26: “mother plants” – please specify the growth conditions.

Line 37: Add keywords that are not included in the title.

Line 63: Instead of “important medicinal plants” add “Medicinal plants of interest”

Line 64: Instead of “modern plant tissue culture techniques called micropropagation” add “micropropagation, a widely used plant tissue culture method”

Line 66: Consider adding: “In addition to rapid propagation and metabolite production, plant tissue culture plays a crucial role in the conservation of valuable and often endangered plant species, such as A. bracteosa” – references for you consideration: https://doi.org/10.3390/plants14101526 https://doi.org/10.3390/horticulturae10010045

Line 66: When clonal propagation is needed, the benefit of ….

Lines 80-81: how about mother plants mentioned in the abstract?

Line 86: verified or characterized?

Line 86: My suggestion is to include his name in the acknowledgements and not here.

Line 86: Please specify how the material was stored before use and how long after collection the in vitro initiation was performed. Add this information to the text

Line 118: "the optimized cytokinin" - optimized? please clarify

The word ‘optimized’ appears with both ‘z’ and ‘s’ spellings throughout the manuscript. Please revise the manuscript to use a consistent spelling.

Line 127: how long was each phase? Add this to the text

Line 129: how old?

Line 131: grammar review: MS medium without auxins was used as a control.

Line 136: add temperature and photoperiod

Lines 136-138: it is not clear for reproduction

Line 149: “with the mother plant” – please specify the growth conditions, age, etc.

Line 150: Describe the abbreviations and the procedures used – there is a reference, but it is important to describe the procedures used

Line 175: same comment from line 149

Line 222: “thrice”?? three times

Line 234: how about contamination? Add this to the text.

Lines 236-239: Is this common? How about reserves in the explant? Has this been observed in other explants of this plant species? Please add this to the discussion

Line 239: In which plant species?

Line 272: same comment from line 222 – revise this throughout the document

Line 346: some workers”??

Line 555: how about initiation?

Revise reference list following journal guidelines

Round 2

Reviewer 3 Report (New Reviewer)

Comments and Suggestions for Authors

I can see an improvement in this version compared to the previous one. I am just suggesting a few minor points that should be addressed before publication:

Line 96 – Please specify the starting material, including its size.

Line 115 – "Agar" is mentioned twice in the same sentence; please revise.

Line 125 – Are the “internodal sections” with or without leaves? Please clarify.

Lines 255–259 – A more thorough discussion is needed for these results, as the referenced study does not use the same type of explants.

Table 4 – Consider changing “% response” to “rooting %” for clarity.

Please also check if the table formatting is consistent with the journal’s style guidelines.

Author Response

We are grateful for the insightful and constructive comments provided by the reviewers and the editorial team. We have carefully addressed all the suggestions and made the necessary revisions to improve the quality and clarity of our manuscript.

Thank you

Sincerely,

Professor Anwar Shahzad

Dr. Zishan Ahmad

Response to the reviewer#3

Comment 1. Line 96 – Please specify the starting material, including its size.

Explanation 1. We appreciate your helpful suggestion. In response, we have clarified the starting material including its size in the Methods section.

Comment 2. Line 115 – "Agar" is mentioned twice in the same sentence; please revise.

Explanation 2. Thank you for pointing that out. We have revised the sentence to remove the repeated mention of the word "Agar".

Comment 3. Line 125 – Are the “internodal sections” with or without leaves? Please clarify.

Explanation 3. The internodal sections used in our experiments were without leaves. We have revised the manuscript to explicitly state this detail to avoid any ambiguity.

Comment 4. Lines 255–259 – A more thorough discussion is needed for these results, as the referenced study does not use the same type of explants.

Explanation 4. Thank you for your valuable comment. We acknowledge that the previously cited study employed a different type of explant, which limited its direct relevance to our findings. In the revised version, we have replaced it with a more suitable reference that corresponds more closely to the explant type used in our study. Furthermore, we have expanded the discussion to provide a more detailed interpretation of the results across all the parameters we examined, ensuring that the context and implications are more clearly articulated.

Comment 5. Table 4 – Consider changing “% response” to “rooting %” for clarity.

Explanation 5. Thank you for pointing that out. We have changed “% response” to “rooting %” as recommended.

Comment 6. Please also check if the table formatting is consistent with the journal’s style guidelines.

Explanation 6.

Thank you for highlighting the importance of consistent table formatting. We have thoroughly reviewed all tables and adjusted them to comply with the journal’s style guidelines, including alignment, font size, and the correct use of units and abbreviations. Additionally, we have used the journal’s provided template for formatting, and we acknowledge that minor formatting issues may still arise during the publication process. However, we are happy to cooperate with the journal’s production team to ensure that any such issues are resolved in the final version.

Comment 7. Figures and tables can be improved.

Explanation 7.

Thank you for the suggestion. We tried to improve the figure quality as much as possible.

This manuscript is a resubmission of an earlier submission. The following is a list of the peer review reports and author responses from that submission.

Round 1

Reviewer 1 Report

Comments and Suggestions for Authors

Please view my comments on the uploaded file (the actual manuscript PDF) in conjunction with the comments below:

The rationale for this study is very weak. There are multiple reasons for this:

  • It is claimed that the species, Ajuga bracteosa ‘cures’ serious diseases such as cancer! Authors are cautioned against making such unvalidated claims. Moreover, when I tried to access the cited reference, the link did not work. I searched for the article on Google Scholar and there was a citation but no accessible publication. So, this was a dubious literature source.
  • There is mention of ‘habitat loss’ and ‘challenging climate’ but no substantiation of the causes and how they are impacting the growth/continuance of this species.
  • There is mention of ‘imminent danger of extinction’ and ‘may become endangered’ – besides these conflicting remarks, I could not find a reliable source to validate this species’ conservation status as being ‘endangered’ or on the brink of extinction. Moreover, I found the following article:

Ali, B.A., Ganie, A.H. and Nawchoo, I.A., 2016. Conservation status of Ajuga bracteosa Wall ex Benth: An important medicinal plant species of Kashmir Himalaya. Int J Ecol Ecosolution, 3(1), pp.1-6. 

The above paper was designed to study the conservation status of Ajuga bracteosa, and it states this species: "does not qualify for any of the threat category in Kashmir valley" – the same area in which the study under review was performed.

  • There are several other publications on the micropropagation of this species. There is no justification as to why there was a need for another publication and why internodes were chosen as explants. The authors even cite one protocol (Ali et al. [no year provided, is it number 13 or 35 in the reference list?] - why was this published protocol not followed to generate materials for the chemical and genetic analyses in the study under review? If the plant has been documented as suitable for tissue culture in other studies, why was there a need to use a different explant for the present study?

The methods are lacking in some important information as appears on the PDF submitted with this report. There are also some treatments/assays with very small sample sizes.

In the discussion, I found a conceptual misunderstanding of what direct organogenesis is.

The authors did not provide an image that clearly showed the internodal segment used as explant. To convince the reader that shoots grew directly on the internodal segment (and not from any meristematic or nodal tissue that may have been inadvertently left behind during explant excision), there must be a representation of the actual explant excised.

The study included a detailed investigation of stomata, however, I found this to be unnecessary as the findings corresponded with obvious growth/physiological responses by in in vitro cultures/plants. The stomata wouldn't really be functional in an artificial environment since carbon is derived via the roots from the sucrose in the medium (and not from the surrounding atmosphere in the culture vessel). I am struggling to understand why the stomata needed to be investigated.

No inferences (or actual quantifications) were made on the quantity of each compound between the regenerants and the mother plant. Documenting the presence of phytocompounds in clones should go beyond qualitative assays; if one wants to consider tissue-cultured materials for mass generation of phytocompounds, the amounts of such then become important. If they are too low, it means then that more effective protocols should be developed, or more plants must be produced from those protocols already established. None of these were even raised in the discussion.

Another important point, what were the yields of regenerated plants in this study? i.e. how many plants can one theoretically and practically obtain from the best shooting/rooting treatments and those plants surviving acclimatization? Only once yields are calculated, can one confidently use the term ‘mass propagation’.

The discussion is weak, mostly simply stating the results and referring to similar studies. Little in-depth engagement on the findings and showing how they can be applied.

The manuscript is largely understandable but has frequent grammatical errors, poor tense usage, awkward phrasing, strange/unusual words in scientific writing, and occasional redundancy. Professional language editing is recommended.

Reviewer 2 Report

Comments and Suggestions for Authors

Dear authors,

The article entitled Micropropagation of Ajuga bracteosa via direct organogenesis using internodal explants: SEM, GC-MS, and SCoT marker analysis is interesting and requires some modifications.

Keywords: SEM and GC-MS are inappropriate.

- On Introduction:

Include a paragraph (or lines) on the importance of genetically stable or somaclonal variation for micropropagation. This justifies the present study.

- On Materials and Methods:

On line 70, included Geographic Localization (India map).

On line 92, item 2.2 is the same as line 102, item 2.3 (Shoot induction and multiplication)

On line 100, what is GLU ?

On line 123, MS I ? Did not cite.

- On Results and Discussion:

Item 3.1 Micropropagation 

  • Lines 198-204 aren't necessary. The results started on: MS basal medium without PGRs...
  • Move Table 4 after item 3.2 in vitro rooting
  • lines 318-322 was not cited in Materials and Methods (move to M and M)
  • lines 322-323: Is it a new result? Did you study different kinds of substrates?
  • lines 329-339: It is not suitable to cite these lines in Results. Cite just the work Results.
  • lines 365-368: It is not suitable to cite these lines in Results. Cite just the work Results.
  • lines 384-389: It is not suitable to cite these lines in Results. Cite just the work Results.
  • The Figures must be in the right position in the text.

Comments on the Quality of English Language

The Quality of English Language can be improved. Many singular and plural words are incorrect.

Reviewer 3 Report

Comments and Suggestions for Authors

Dear Author;

In the manuscript, the averages in the text and "± ME or ± SD" values ​​should not be given. It can be given in the table, but it is not necessary in the sentence.

Lines 25-29: SEM, GC-MS, and Scot analysis should be given important results.

Line 92 and 102: The title “Shoot Induction and Multiplication” has been used twice, one should be removed.

Line 115–126: Instead of the expression “Thermocol Cups ,, a more technical term such as“ Polyethylene Pots ”should be preferred.

Line 129–135: PCR conditions are not detailed enough. For example: PCR loop protocol (Denaturation, Handaling, Extension times) missing.

Line 136–144: In the SEM procedure, the duration and concentration of each step should be given clearly instead of “Alcohol Series”.

Line 145–165: GC-MS device settings contain more technical details, but the extraction method (eg: soxhlet, ultrasonic, etc.) is not specified.

Line 187: “2.9 8.Statistical Analysis” expression is the spelling error. It should be corrected as “2.9 statistical analysis ..

Line 193: "P" should be lowercase and italic.

Line 197: "A. bracteosa" must be italic. The entire manuscript has plant species that are not written in Italic. should be checked.

Line 228: "P ≤ 0.05" instead of "(p> 0.05)" in Table 1 and other tables. "P" must be lowercase and italic.

Line 536: “minimal danger of genetic instability” is overly ambitious. Monomorphic bands alone are not sufficient to exclude somaclonal variation.

Line 515: The expression "(P=0.05)" is incorrect for not significantly different. It should be (P > 0.05).

SCoT (Start Codon Targeted) markers generally target conserved regions close to the ATG translational start codon region in the plant genome. Therefore, they are suitable for detecting genetic diversity in gene-coding or nearby regions. However: Transposon regions are usually located in repetitive, epigenetically suppressed heterochromatic regions. It is not possible for SCoT markers to directly target these regions because primer design is usually directed at gene starts. Therefore, detection of variations due to mobile elements or epigenetic/retrotransposon-based polymorphisms with SCoT markers is quite limited. The use of iPBS (inter-Primer Binding Site) markers would have been more meaningful in this study. iPBS markers are more sensitive for the detection of somaclonal variation because retrotransposon activation may increase in stressful environments such as tissue culture. It would be stronger to prefer retrotransposon-based systems such as iPBS or IRAP instead of a gene-focused marker such as SCoT. Obtaining monomorphic bands with SCoT does not mean that there is no variation. It may just show that the markers used cannot detect that variation, even if it exists.

Therefore, I do not classify adding additional analyses to a study that has been studied and completed, but if possible, adding analyses with iPBS or IRAP markers will increase the power of the study.